# The role of pulmonary rehabilitation in idiopathic pulmonary fibrosis: An overview of systematic reviews

**Shangyue Song**[1,2], **Zhenzhen Feng**[1,2,3], **Wenrui Liu**[1,2], **Jiansheng Li**[1,2,3]*

**1** The First Clinical Medical College, Henan University of Chinese Medicine, Zhengzhou, Henan, People's Republic of China, **2** Co-Construction Collaborative Innovation Center for Chinese Medicine and Respiratory Diseases by Henan & Education Ministry of P.R. China, Henan University of Chinese Medicine, Zhengzhou, Henan, People's Republic of China, **3** Department of Respiratory Diseases, The First Affiliated Hospital of Henan University of Chinese Medicine, Zhengzhou, Henan, People's Republic of China

* li_js8@163.com

## Abstract

### Background

The role of pulmonary rehabilitation (PR) in idiopathic pulmonary fibrosis (IPF) has been studied in several systematic reviews (SRs), but no definitive conclusions have been drawn due to the wide variation in the quality and outcomes of the studies. And there are no studies to assess the quality of relevant published SRs. This overview aims to determine the effectiveness of PR in patients with IPF and to summarize and critically evaluate the risk of bias, methodological, and evidence quality of SRs on this related topic.

### Methods

With no language restrictions, eight databases were searched from inception to March 10, 2023. The literature search, screening, and data extraction were carried out separately by two reviewers. We assessed the risk of bias using the ROBIS tool, the reporting quality using PRISMA statements, the methodological quality using AMSTAR-2, and the evidence quality using Grades of Recommendations, Assessment, Development, and Evaluation (GRADE).

### Results

Seven SRs from 2018–2023 (including 1836 participants) on PR for the treatment of IPF were selected, all of which included patients with a definitive diagnosis of IPF. After strict evaluation by the ROBIS tool and AMSTAR-2 tool, 42.86% of the SRs had a high risk of bias and 85.71% of the SRs had critically low methodological quality in this overview. PR might be effective for patients with IPF on exercise capacity, quality of life, and pulmonary function-related outcomes, but we did not find high quality evidence to confirm the effectiveness.

### Conclusion

PR may appear to be an effective and safe treatment for patients with IPF, but the results of this overview should be interpreted dialectically and with caution. Further high-quality,

**Data Availability Statement:** All relevant data are within the paper and its Supporting Information files.

**Funding:** The author(s) received no specific funding for this work.

**Competing interests:** This work was supported by JS L: the Chinese Medicine Prevention and Treatment of Respiratory Diseases National the Chinese Medicine Inheritance and Innovation team (ZYYCXTD-C-202206); JS L: the Chinese Medicine Inheritance and Innovation "Hundred and Ten Million" Talent Project - Chief Scientist of Qi-Huang project [(2020) NO. 219]; ZZ F: Establishment of diagnostic criteria of TCM syndromes for two respiratory diseases, including lung cancer (STG-ZYX03-202123). The funders had no role in study design, data collection and analysis, decision to publish, or preparation of the manuscript.

rigorous studies are urgently needed to draw definitive conclusions and provide scientific evidence.

## Introduction

IPF is a chronic, fibrosing interstitial pneumonia of unknown cause that etiology characterized by dry cough, fatigue, dyspnea, and progressive deterioration of lung function [1,2]. It is commonly occurring in the elderly, with an incidence of approximately 0.33–4.51 per 10,000 [3]. Patients with IPF have a poor prognosis, with a median survival of approximately 3–5 years after diagnosis [4], and acute exacerbation can lead to respiratory failure or even death [5]. The current treatments for IPF are mainly pharmacological (pirfenidone, nintedanib) and non-pharmacological [6–9]. Pharmacological treatments can alleviate some of the decline in lung function, but the side effects and high cost of medications can reduce the quality of life for patients and place a significant financial burden on individuals [10–12].

PR is an essential component of the non-pharmacological management of IPF. It is a comprehensive intervention based on a thorough patient assessment followed by a tailored treatment design to improve the physical and psychological status of patients with chronic respiratory disease and to promote long-term adherence to health-promoting behaviors [13]. Studies have shown that it has a positive effect on the quality of life and functional status of patients with IPF and is the only intervention that improves exercise tolerance [14–17].

In recent years, PR has been better applied to chronic respiratory diseases such as chronic obstructive pulmonary disease (COPD) [18,19]. However relatively few studies have been conducted on the effects of PR in IPF. In addition, the quality of the studies has been variable, posing a significant challenge for clinical decision-making. Therefore, there is a strong need to evaluate the current evidence on the effectiveness and safety of PR in the treatment of IPF.

This is the first overview to comprehensively evaluate SRs for PR in IPF patients. The aim of this overview is to critically evaluate the quality of the relevant SRs and to provide a comprehensive and objective assessment of efficacy and safety to strengthen the evidence base for clinicians.

## Methods

### Protocol and registration

The overview protocol was registered on the International Prospective Register of Systematic Reviews (*The role of pulmonary rehabilitation in idiopathic pulmonary fibrosis*: *an overview of systematic reviews*, CRD42023407753) and conducted by the Preferred Reporting Items for Systematic Reviews and Meta-Analyses (PRISMA) guidelines. No ethical application was required.

### Search strategy

International electronic databases (PubMed, Cochrane Library, Embase, Web of Science) and Chinese electronic databases (Chinese National Knowledge Infrastructure, Wan Fang database, Chinese biomedical literature service system, and Chongqing VIP) were systematically searched from their inception to March 10, 2023, with no restrictions on language. The search terms were mainly used as Mesh terms combined with free words. Keywords include Idiopathic Pulmonary Fibrosis, Pulmonary Fibrosis, Idiopathic, IPF, Pulmonary Fibrosis, Lung Disease, Interstitial, PR, Rehabilitation Therapy, Rehabilitation Training, Rehabilitation

Programs, Exercise Therapy, Respiratory Exercise, Meta-Analysis, Systematic Review, and so on. Detailed search strategies are shown in the S1 Table. We also review the references in the included literature and the relevant systematic reviews.

## Inclusion and exclusion criteria

We included studies that matched the following criteria:

Types of Studies: SRs included randomized controlled trials (RCTs) of PR for patients with IPF, with No limitation on language.

Types of Participants: Participants were diagnosed with IPF according to national or international clinical guidelines, with no age, gender, or national origin restrictions.

Types of Interventions: The Intervention group included PR alone (PR was defined as a comprehensive program composed at least one of exercise training, educational lectures, or self-administered) or combined with the other treatments (Conventional treatment). The control group Intervention received conventional treatment (such as vital sign testing and dietary care), placebo, blank control, or other treatments (conventional medical treatments or other nondrug treatments).

Types of Outcome Measures: The outcomes of interest were as follows:(1) Exercise tolerance:6minute walking test(6MWD); (2) Quality of life: St. George's Respiratory Questionnaire /IPF-specific St. George's Respiratory Questionnaire (SGRQ/SGRQ-I); (3) Pulmonary function: FVC% pred (forced vital capacity percent predicted), DLCO% (diffusing capacity for carbon monoxide); (4) Dyspnea score: MRC/mMRC (Medical Research Council scores /modified British Medical Research Council scores); (5) Adverse Effects.

The following articles were excluded: (1) duplicated publications; (2) updated SRs; (3) network meta-analysis; (4) conference abstracts or systematic reviews protocols; (5) studies on which the data could not be extracted.

## Literature screening and data extraction

Two researchers (SY Song and WR Liu) independently conducted the literature search, screening based on the inclusion and exclusion criteria. Literature management was conducted using EndnoteX9. Duplicates were removed, titles and abstracts of articles were read to exclude those that did not meet the criteria, and articles with potential inclusion were then read in full to select those for final inclusion.

One researcher (SY Song) extracted the following basic information: first author, publication date, sample size, country, interventions, comparators, outcome measures, methodological quality assessment, and conclusion. Another researcher (ZZ Feng) checked the extracted data, and if there was a difference of opinion, referred to the original text and revised it.

## Strategy for data synthesis

We provide a narrative description of the summary statistics from the included reviews. The findings are reported in text and tabular form, accompanied by an evaluation of the quality of the evidence. The data were categorized based on the objectives of the intervention and the types of outcomes reported.

## Assessment of included SRs

The quality of the included studies was independently assessed and cross-checked by the two researchers (SY Song and WR Liu). Any disputes were resolved by negotiation or by a third researcher (ZZ Feng) acting as an arbitrator.

**Risk of bias evaluation.** The ROBIS tool aims to evaluate the level of bias presented in a systematic review. This bias assessment tool covers 3 phases: (1) assessing relevance (optional according to the situation); (2) Identifying concerns with the review process (study eligibility criteria, identification and selection of studies, data collection and study appraisal, synthesis, and findings); (3) Judging risk of bias. The results were rated as "high risk", "low risk" or "unclear risk" [20].

**Report quality evaluation.** The PRISMA statement [21] was used to assess the quality of reporting in the included literature. The checklist consists of 27 items in 7 areas for a total score of 27. Each item was scored based on the standardization and completeness of the report, with 1 point indicating full compliance, 0.5 points indicating partial compliance, and 0 points indicating no report. Reports with a score of >21–27 were considered to be relatively complete, those with a score of 15–21 were considered to be reports with some deficiencies, and ≤15 were considered to be reports with a relatively serious lack of information.

**Methodological quality evaluation.** The AMSTAR-2 was used to assess the methodological quality of the included studies and contains 16 items, of which 7 items are critical (items 2, 4, 7, 9, 11, 13, 15) [22]. Depending on the degree of compliance, each item is described as 'Yes', 'Partial Yes', or 'No'. Finally, the methodological quality is divided into four-level: high, moderate, low, and critically low.0-1 non-critical item nonconformity is considered as high quality, >1 non-critical item nonconformity is considered as moderate quality, 1 critical item nonconformity (with or without non-critical item nonconformity) is considered as low quality, >1 critical item nonconformity is considered as critically low quality regardless of non-critical item nonconformity.

**Evidence quality evaluation.** The GRADE system was used to evaluate the quality of primary evidence. Five degrading factors were as follows: Limitations, Inconsistency, Indirectness, Imprecision, and Publication Bias [23] and the evidence quality was rated as high, moderate, low, and critically low. RCTs are high-quality evidence, downgrade 1 is moderate, downgrade 2 is low and downgrade 3 is critically low.

## Results

### Literature search and selection

446 documents were retrieved from the 8 databases. After 93 duplicates were excluded, 353 records were screened by reading the titles and abstracts, and 321 doing the full texts, 25 publications were excluded. Finally, 7 studies [24–30] were included after an independent review by two researchers. The study selection process and reasons for exclusion are detailed in Fig 1 below.

### Characteristics of included SRs

The characteristic of the 7 SRs were summarized in Table 1. We included 7 SRs, [24–30] with 3 studies [24–26] published in Chinese and 4 studies [27–30] published in English, all published between 2018 and 2023. The RCTs included in each study ranged from 4 to 9, and the overall number of study subjects ranged from 138 to 549 cases. The study participants were all patients with IPF, and the intervention group intervention measures were mainly PR or PR with conventional treatment, while the treatments in the control group mainly consisted of conventional treatment or non-PR. In terms of quality assessment tools, 6 studies [24–28,30] used the Cochrane risk of bias assessment criteria, and 1 study [29] used the PEDro scale to assess the quality of the included studies.

### Risk of bias of included SRs

Concerning the ROBIS tool, phrase 1 can be selected contextually and was not implemented in our study. Domain 1 was used to assess the eligibility criteria for the primary study and

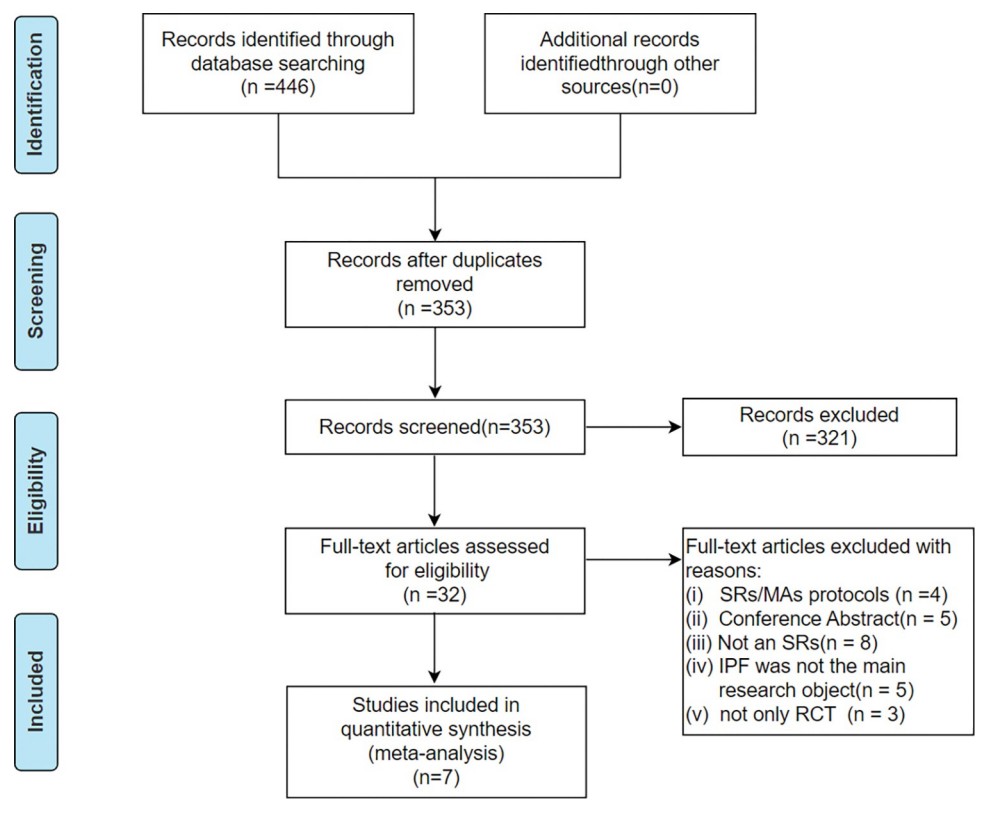

**Fig 1. Flowchart of literature screening.**

whether there was evidence that the objectives and eligibility criteria were pre-specified, with a low risk of bias for 7 SRs [24–30] (100%). Domain 2 was used to evaluate the identification and selection of studies, and 4 SRs [27–30] (57.14%) were rated as low risk of bias.5 SRs (71.43%) [25,27–30] with a low risk of bias and 2 SRs [24,26] (28.57%) with an unclear risk of bias in Domain 3, which assessed data collection and study appraisal. Domain 4 was used to assess synthesis and findings, 1 SRs [30] (14.29%) was rated as a low risk of bias, while the remaining 6 SRs [24–29] were all rated as high risk of bias. Stage 3 assessed the overall risk of bias in the review, with 4 SRs [27–30] (57.14%) having a low risk of bias. The results of the ROBIS tool evaluation are shown in S2 Table and S1 Fig.

## Report quality of included SRs

The PRISMA checklist shows an average score of 17.92 for the included studies, with total scores ranging from 14.5 to 24. 1 SRs [24] scored <15, 3 SRs [25,26,29] scored between 15–21, and 2 SRs [28,30] ≥21. The results show that titles, theoretical basis, purpose, sources of information, data entries, combined effect indicators, study characteristics, individual outcomes, and descriptions of statistical analyses were well reported for all studies (100%). However, some reporting deficiencies remain structured summaries, inclusion criteria, searches, data extraction, synthesis of results, other analyses, etc. Reporting deficiencies were mainly found in the following items: evidence quality assessment methodology (14.29%), reporting bias (14.29%), evidence quality assessment (14.29%), registration (28.57%), funding (42.86%) and disclosure (0%). S3 Table presents the overview of PRISMA checklist items.

**Table 1. Characteristics of the included SRs.**

| Author (Year) | Trials (n) | Country | Interventions Group | Control Group | Outcomes | Methodological Quality Assessment | Conclusion |
|---|---|---|---|---|---|---|---|
| Guo et al,2023 [24] | 9(373) | China | PR +Conventional treatment and care | Conventional treatment and care | Exercise tolerance (6MWD); Pulmonary function(FVC,DLCO%); Peak heart rate; Quality of life(SGRQ); Adverse events | Cochrane Collaboration's Risk of Bias tool | The evidence suggests that pulmonary rehabilitation can improve exercise capacity, cardiopulmonary function and quality of life in patients with idiopathic pulmonary fibrosis. |
| Fu et al,2021 [25] | 7(211) | China | PR +Conventional treatment | Usual care | Exercise tolerance (6MWD); Pulmonary function(FVC% pred); Quality of life(SGRQ-I) | Cochrane Collaboration's Risk of Bias tool | Pulmonary rehabilitation is effective in improving exercise tolerance, pulmonary function indexes, FVC% pred, and quality of life in patients with idiopathic pulmonary fibrosis. |
| Cheng et al,2019 [26] | 6(233) | China | PR | Non-PR | 6MWD; Oxygen consumption (VO$_2$); Dyspnea(MRC); Quality of life(SGRQ); Adverse events | Cochrane Collaboration's Risk of Bias tool | Pulmonary rehabilitation is beneficial for short-term regression of IPF patients, whether it is beneficial for long-term regression requires further sample size expansion. |
| Cheng et al,2018 [27] | 4(142) | China | PR | Once weekly telephone calls, Regular medical care,Normal physical activity | Long-term/Short-term effect on exercise capacity (6MWD); Long-term/Short-term effect on quality of life (SGRQ)/SGRQ-I | Cochrane Collaboration's Risk of Bias tool | In patients with IPF, pulmonary rehabilitation showed short-term effects in enhancing exercise capacity and health-related quality of life, while it had no detectable effects at long-term follow-up. |
| Yu et al,2019 [28] | 7(190) | China | PR + Regular care | Regular care | Exercise capacity (6MWD); Lung function (FVC%, DLCO%); Quality of life (SGRQ/ SGRQ-I); Adverse events | Cochrane Collaboration's Risk of Bias tool | This study suggests that PR may enhance exercise capacity and improve quality of life in IPF patients. Besides, PR may also delay the decline of lung function of patients with IPF. |
| Mansueto et al,2018 [29] | 5(138) | Brazil | PR | Usual care | Exercise tolerance (6MWD); Quality of life (SGRQ) | PEDro scale | Pulmonary rehabilitation is effective in increasing exercise tolerance and improving quality of life in patients with idiopathic pulmonary fi brosis. |
| Lei et al,2022 [30] | 11 (549) | China | PR | Usual care | Exercise tolerance (6MWD); Quality of life (SGRQ)/SGRQ-I); Pulmonary function (FEV% pred,DLCO%); Dyspnea (mMRC) | Cochrane Collaboration's Risk of Bias tool and the Grading of Recommendations, Assessment, Development and Evaluation criteria. | Pulmonary rehabilitation may significantly improve exercise tolerance and quality of life in idiopathic pulmonary fibrosis patients, but the quality of evidence was low to moderate. |

## Methodological quality of included SRs

The AMSTAR-2 results are shown in Fig 2. 1 SRs [30] was considered low quality and 6 SRs [24–30] were considered critically low quality. Methodological limitations arose from the following critical items: item 2 (only 2 studies reported their study protocols), item 13 (no studies stated whether the risk of bias in the included studies was considered in the outcome), and item 15 (only 1 study assessed publication bias and discussed the impact of bias). The following non-critical items also affected the methodological quality of the included studies: item 3 (all studies did not explain the type of study included), item 10 (no study reported the source of funding for the included studies), and item 12 (no study assessed the impact of the risk of bias of the included studies on the meta-analysis).

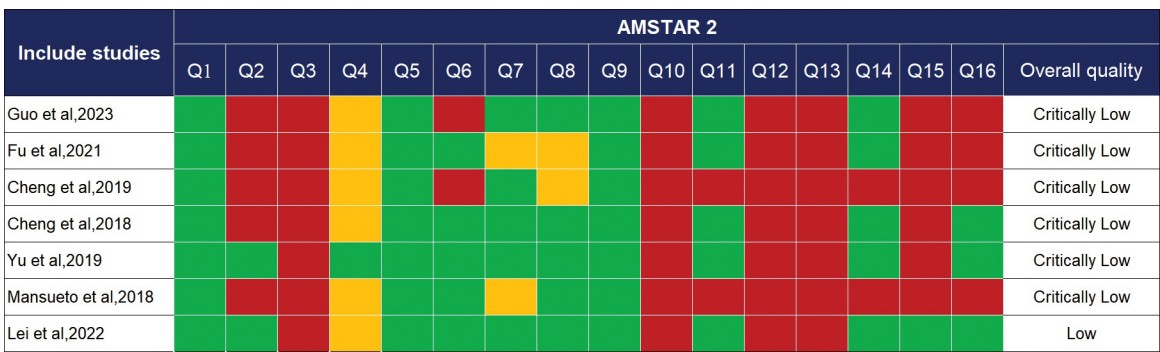

**Fig 2. Critical appraisal of studies included using the AMSTAR 2 tool.** Q1. Did the research questions and inclusion criteria for the review include the components of PICO? Q2. Did the report of the review contain an explicit statement that the review methods were established before the conduct of the review and did the report justify any significant deviations from the protocol? Q3. Did the review authors explain their selection of the study designs for inclusion in the review? Q4. Did the review authors use a comprehensive literature search strategy? Q 5. Did the review authors perform study selection in duplicate? Q6. Did the review authors perform data extraction in duplicate? Q7. Did the review authors provide a list of excluded studies and justify the exclusions? Q8. Did the review authors describe the included studies in adequate detail? Q9. Did the review authors use a satisfactory technique for assessing the risk of bias (RoB) in individual studies that were included in the review? Q10. Did the review authors report on the sources of funding for the studies included in the review? Q11. If meta-analysis was performed did the review authors use appropriate methods for statistical combination of results? Q12. If meta-analysis was performed, did the review authors assess the potential impact of RoB in individual studies on the results of the meta-analysis or other evidence synthesis? Q13. Did the review authors account for RoB in individual studies when interpreting/ discussing the results of the review? Q14. Did the review authors provide a satisfactory explanation for, and discussion of, any heterogeneity observed in the results of the review? Q15. If they performed quantitative synthesis did the review authors carry out aeived for conducting the review? figure color: Green, Yes; Yellow, Partial Yes; Red, No.

## Effectiveness of PR and the evidence quality

Descriptive analysis was used to evaluate the outcomes of interest in our study, as quantitative analysis was not feasible with the obtained data.

29 outcomes from 7 studies were assessed using the GRADE system, of which 2 (6.90%) were of moderate quality, 12 (41. 83%) were of low quality, and 15 (51.72%) were of critically low quality, with no high quality outcomes identified. Limitations (100%) and Imprecision (86.21%) were the two most significant factors reducing the quality of evidence, followed by publication bias (51.72%) and inconsistency (27.59%). The specific results are presented in Fig 3 and Table 2.

**Exercise capacity.** All 7 studies [24–30] used the 6MWD to assess the effect of PR on exercise capacity in patients with IPF. The results of the meta-analysis showed that PR increased the 6MWD distance and significantly improved exercise capacity in IPF.1 Study [27] showed no significant difference between PR effects on 6MWD in the short and long term. The quality of evidence for 1 [30] outcome was moderate, 1 [28] outcome was low and 6 [24–27,29] outcomes were critically low.

**Quality of life.** 6 studies [24,25,27–30] reported quality of life using the SGRQ/SGRQ-I scores.6 studies [24,25,27–30] pooled data for meta-analysis and 1 study [26] presented comprehensive descriptive results, all showing that PR improves the quality of life in patients with IPF. The quality of evidence for 1 outcome [30] was moderate, 4 outcomes [24,27–29] were low quality, and 3 outcomes [25,27,29] were critically low quality.

**Pulmonary function.** The main pulmonary function outcome measures were FVC, FVC % pred, and DLCO%. Except for 2 studies [24,28] in which the outcomes of DCLO% were not statistically significant, the outcome indicators of the other studies [24–26,28,30] showed that PR can significantly improve pulmonary function in patients with IPF.2 studies [24,28] used the outcome FVC%, all with low quality; 2 studies [25,30] used the outcome FVC% pred, 1

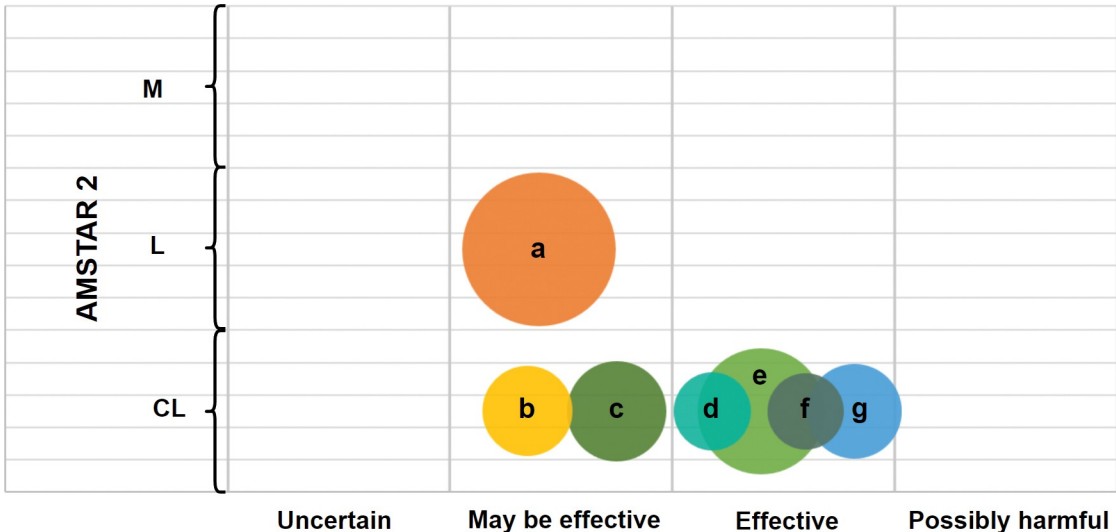

**Fig 3. Evidence mapping of the literature on systematic evaluation of IPF for pulmonary rehabilitation.** Bubbles size: sample size; X-axis: Author conclusion (Uncertain, May be effective, Effective, possibly harmful); Y-axis: AMSTAR 2 score; a, Lei et al,2022 [30]; b, Yu et al,2019 [28]; c, Cheng et al,2019 [26]; d, Cheng et al,2018 [27]; e, Guo et al,2023 [24]; f, Mansueto et al,2018 [29]; g, Fu et al,2021 [25]; CL, critically low; L, low; M: moderate.

study [30] had low quality and 1 study [25] had very low quality; 3 studies [24,27,30] used the outcome DLCO%, all had low quality.

**Dyspnea.** 2 studies [26,30] used MRC/mMRC to assess the effect of PR on dyspnea in patients. 1 SRs [26] result was not statistically significant, while the other SRs [30] showed that PR significantly improved dyspnea. The quality of both pieces of evidence is very low.

**Oxygen consumption.** 1 study [26] reported changes in $VO_2$ in IPF patients after treatment. Meta-analysis results showed that PR increased $VO_2$ in IPF patients with a statistically significant difference.

**Adverse effects.** Adverse events were tracked in 4 [24,26,28,30] of the 7 included studies. However, in terms of outcomes, adverse events during PR were relatively rare. 3 of 4 studies [24,26,28] found no adverse events associated with the intervention, and only 1 study [26] reported 4 serious events, but these were considered to be related to acute exacerbations of IPF and not associated with the PR intervention.

## Discussion

### Summary of main results

This study provides a comprehensive evaluation and descriptive analysis of the quality of 7 SRs in patients with IPF treated with PR. The available evidence supports PR as an effective and safe treatment for patients with IPF. Based on the ROBIS tool, 4 SRs were assessed as having a low risk of bias. Using the PRISMA checklist, we found 2 studies to be relatively complete. In terms of methodological quality, all 7 SRs were rated low or critically low.

### Discussion based on the results

According to the ROBIS results, the vast majority of SRs (85.71%) had a high risk of bias in data synthesis and presentation of results. The included studies pooled the data as much as possible. However, none of the trials performed a funnel plot analysis. Therefore, systematic review researchers should conduct a bias analysis of the results section to determine whether

**Table 2. Quality of evidence in included SRs with GRADE.**

| Author(Year) | Outcomes | Trials (n) | 95%CI | Limitations | Inconsistency | Indirectness | Imprecision | Publication Bias | Quality of Evidence |
|---|---|---|---|---|---|---|---|---|---|
| Guo et al,2023 [24] | 6MWD | 9(373) | 30.48 [17.46,43.50] | -1[a] | -1[b] | 0 | -1[c] | 0 | Very Low |
| | FVC | 6(248) | 5.60[3.80,7.40] | -1[a] | 0 | 0 | -1[c] | 0 | Low |
| | DLCO% | 4(160) | 2.13[-0.24,4.51] | -1[a] | 0 | 0 | -1[c] | 0 | Low |
| | Peak heart rate | 2(123) | 2.11[-0.50,4.72] | -1[a] | 0 | 0 | -1[c] | 0 | Low |
| | SGRQ | 6(271) | -9.16[-10.44,-7.88] | -1[a] | 0 | 0 | -1[c] | 0 | Low |
| Fu et al,2021 [25] | 6MWD | 6(190) | 35.32 [22.65,48.00] | -1[a] | 0 | 0 | -1[c] | -1[d] | Very Low |
| | FVC%pred | 3(90) | 5.26[1.67,8.85] | -1[a] | 0 | 0 | -1[c] | -1[d] | Very Low |
| | SGRQ-I | 4(109) | -9.06[-12.02,-6.10] | -1[a] | 0 | 0 | -1[c] | -1[d] | Very Low |
| Cheng et al,2019 [26] | 6MWD | 6(198) | 39.21 [17.69,60.74] | -1[a] | -1[b] | 0 | -1[c] | -1[d] | Very Low |
| | $VO_2$ | 3(87) | 1.59[0.80,2.37] | -1[a] | -1[b] | 0 | -1[c] | -1[d] | Very Low |
| | MRC | 5(175) | -0.35[-0.74,0.04] | -1[a] | -1[b] | 0 | -1[c] | -1[d] | Very Low |
| Cheng et al,2018 [27] | 6MWD(St) | 4(142) | 38.38[4.64,72.12] | -1[a] | -1[b] | 0 | -1[c] | -1[d] | Very Low |
| | SGRQ/SGRQ-I (St) | 3(121) | -8.40 [-11.44,-5.36] | -1[a] | 0 | 0 | -1[c] | -1[d] | Low |
| | 6MWD(Lt) | 2(93) | 17.02 [-26.87,60.81] | -1[a] | 0 | 0 | -1[c] | -1[d] | Very Low |
| | SGRQ/SGRQ-I (Lt) | 2(93) | -3.45[-8.55,1.64] | -1[a] | 0 | 0 | -1[c] | -1[d] | Very Low |
| Yu et al,2019 [28] | 6MWD | 6(169) | 48.6[29.03,68.18] | -1[a] | 0 | 0 | -1[c] | 0 | Low |
| | SGRQ/SGRQ-I | 4(113) | -7 .87 [-11.44,-4.30] | -1[a] | 0 | 0 | -1[c] | 0 | Low |
| | FVC% | 3(90) | 3.69[0.16,7 .23] | -1[a] | 0 | 0 | -1[c] | 0 | Low |
| | DLCO% | 3(90) | 3.02[-0.38, 6.42] | -1[a] | 0 | 0 | -1[c] | 0 | Low |
| Mansueto et al,2018 [29] | 6MWD | 4(113) | 44.01[5.26,82.76] | -1[a] | -1[b] | 0 | -1[c] | -1[d] | Very Low |
| | SGRQ(syptom) | 4(113) | -18.9[-26.88,-10.10] | -1[a] | 0 | 0 | -1[c] | -1[d] | Very Low |
| | SGRQ(activity) | 3(92) | -1.25[-2.60,0.11] | -1[a] | 0 | 0 | -1[c] | -1[d] | Very Low |
| | SGRQ(impact) | 3(92) | -8.97[-11.57,-6.36] | -1[a] | 0 | 0 | -1[c] | -1[d] | Very Low |
| | SGRQ(total) | 3(386) | -7.39[-10.68,-4.09] | -1[a] | 0 | 0 | 0 | -1[d] | Low |
| Lei et al,2022 [30] | 6MWD | 10(447) | 35.15 [25.40,44.90] | -1[a] | 0 | 0 | 0 | 0 | MODERATE |
| | SGRQ/SGRQ-I | 6(303) | -9.14[-10.87,-7.40] | -1[a] | 0 | 0 | 0 | 0 | MODERATE |
| | mMRC | 3(196) | -0.76[-1.25,-0.27] | -1[a] | -1[b] | 0 | -1[c] | 0 | Very Low |
| | FVC% pred | 4(214) | 4.88[2.67,7.10] | -1[a] | 0 | 0 | -1[c] | 0 | Low |
| | DLCO% | 6(358) | 4.71[0.96,8.46] | -1[a] | -1[b] | 0 | 0 | 0 | Low |

a The design of the experiment with a large bias in random, distributive hiding or blind

b $I^2$ is larger

c Confidence interval is not narrow enough; The optimal sample size was not enough

d Fewer studies are included and there may be greater publication bias.6MWD (St): 6MWD (short-term); 6MWD (Lt): 6-WMD (long-term); SGRQ /SGRQ-I(St): SGRQ/SGRQ-I(short-term); SGRQ/SGRQ-I(Lt): SGRQ/SGRQ-I(long-term).

there is publication bias and selective reporting. Second, some systematic reviews may miss studies that meet the inclusion criteria. Thus, researchers should use a complete and detailed search strategy and multiple search methods (For example citation searches, expert contacts, retrospective references, and manual searches). Literature search, screening, and data analysis should be carried out by at least 2 researchers.

The results of the PRISMA and AMSTAR-2 assessments indicate that there is still room for improvement in the quality of reporting and the overall methodological quality of the included SRs. The majority (71.43%) of trials did not register their protocols in advance, which increased the risk of bias in the trials. Authors reporting their study plans prior to registration can reduce the likelihood of bias in the review and improve the transparency of the trial [31]. The included studies had varying degrees of deficiencies in grey literature searches, rational interpretation of risk of bias and heterogeneity, assessment of publication bias, assessment of the quality of impact evidence, and funding, which reduced the quality of the studies. For this reason, researchers should pay close attention to PRISMA and AMSTAR-2 in Future Studies.

The GRADE score indicates that the overall quality of the evidence for the inclusion of SRs is low. All outcomes were downgraded for limitations. This indicates a high risk of original trial bias for the included SRs. The methodological quality assessment of the original RCTs showed that the majority of participants and doctors were not blinded due to the inherent nature of the study methodology; at the same time, there was some risk of bias from partial allocation concealment. Imprecision was also a very common downgrading factor in the included SRs, mainly related to insufficient sample sizes and wide confidence intervals in the included studies [32,33]. In addition, publication bias was mainly reflected in the lack of narrow confidence intervals and the inclusion of studies that did not meet the sample size estimates for clinical trials or studies with potential publication bias. Therefore, more large-sample, multicenter, high-quality RCTs are needed to provide more realistic and objective clinical evidence.

Finally, a synthesis of the available research evidence suggests that PR has some efficacy for patients with IPF. However, based on the results of the assessment, there are some methodological problems with the included SRs. Double-blinding is not feasible in PR, and this problem affects the methodological quality of the evidence. Furthermore, while the GRADE methodology provides a systematic approach to assessing the quality of evidence, there is uncertainty about the best way to implement the methodology in the overview [31]. Therefore, researchers need to consider the characteristics of PR and integrate them with clinical RCTs to find better solutions. In addition, given the overlap between systematic reviews and the clinical heterogeneity of the main studies, this review only provides a summary of the current evidence profile and it is difficult to carry out further systematic analyses of the relevant data.

## Implications for future practice and research

Our evidence indicates that PR may be an effective complementary therapy in the treatment of IPF and is beneficial in improving lung function, exercise tolerance, and quality of life in IPF patients, although the quality of evidence and methodology is low. Based on these findings, future research should focus on the following points: (1) Clinical researchers should strictly adhere to the Uniform Standards for Reporting Clinical Trials (CONSORT) [34] and the Standards for Reporting Interventions in Clinical Trials (STRICTA 2010) to improve the quality of research and the reliability of clinical practice;(2) Systematic reviewers should register the study plan before the start of the study to avoid potential risks of bias;(3) Most clinical practices use a combination of PR to treat IPF; therefore, future comparative studies of specific PR interventions are needed to identify the safest and most effective of these interventions;(4) The

long-term effects of PR in patients with IPF are unclear, and longer follow-up studies are needed to determine the long-term clinical effectiveness of PR.

## Strengths and limitations

To our knowledge, this is the first overview to examine the effectiveness and safety of PR in patients with IPF and could provide a comprehensive evidence base for clinical decision-making. In addition, this study identified the limitations of the included studies and provided targeted recommendations by assessing the included SRs using ROBIS, PRISMA, AMSTAR-2, and GRADE, which may be useful to guide future studies.

Admittedly, there are some limitations to this overview. Firstly, the assessment of included studies can be subjective, and although our overview was assessed independently by 2 researchers, it may still be subject to subjective factors in the assessment process. Secondly, there may be some selection bias because the study did not search the relevant grey literature. Thirdly, this review only narratively summarizes the findings and does not provide a quantitative analysis, as there is considerable clinical heterogeneity between the different systematic reviews in terms of interventions (type, intensity, duration), outcomes, etc. Therefore, the results should still be interpreted with caution.

## Conclusion

What is the precise role of PR in the treatment of IPF? Based on the trials we included, definitive conclusions to this question are difficult to draw. It is unclear whether this is related to the quality of the original studies and the variable quality of the evidence on the results (ranging from very low to moderate). Therefore, considering the limitations of the overview, further scientific research is warranted to investigate the efficacy of PR in IPF, to provide a more robust, scientifically rigorous, and accurate foundation for clinical decision-making.

## Supporting information

**S1 Checklist. PRISMA 2020 checklist.**
(DOC)

**S1 Fig. Graphical presentation of risk of bias of included SRs.**
(TIF)

**S1 Table. Details of the literature search strategy.**
(DOCX)

**S2 Table. Risk of bias of included SRs.** ☺, Low; ☹, High;?, Unclear.
(DOCX)

**S3 Table. Report quality of included SRs.** 1, Guo et al,2023 [24]; 2, Fu et al,2021 [25]; 3, Cheng et al,2019 [26]; 4, Cheng et al,2018 [27]; 5, Yu et al,2019 [28]; 6, Mansueto et al,2018 [29]; 7, Lei et al,2022 [30].
(DOCX)

## Acknowledgments

The authors sincerely thank the faculty and students of Henan University of Traditional Chinese Medicine for their assistance.

## Author Contributions

**Conceptualization:** Jiansheng Li.

**Formal analysis:** Shangyue Song, Zhenzhen Feng, Wenrui Liu.

**Methodology:** Zhenzhen Feng.

**Writing – original draft:** Shangyue Song.

**Writing – review & editing:** Zhenzhen Feng, Jiansheng Li.

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
