## [Decision Letter · Decision Letter 0]

24 Oct 2023

PONE-D-23-30793The role of pulmonary rehabilitation in idiopathic pulmonary fibrosis: an overview of systematic reviewsPLOS ONE

Dear Dr. Li,

Thank you for submitting your manuscript to PLOS ONE. After careful consideration, we feel that it has merit but does not fully meet PLOS ONE’s publication criteria as it currently stands. Therefore, we invite you to submit a revised version of the manuscript that addresses the points raised during the review process.

We look forward to receiving your revised manuscript.

Kind regards,

Felix Bongomin, MB ChB, MSc, MMed, FECMM

Academic Editor

PLOS ONE

Journal Requirements:

Reviewers' comments:

Reviewer's Responses to Questions

**Comments to the Author**

1. Is the manuscript technically sound, and do the data support the conclusions?

Reviewer #1: Yes

2. Has the statistical analysis been performed appropriately and rigorously? 

Reviewer #1: N/A

3. Have the authors made all data underlying the findings in their manuscript fully available?

Reviewer #1: Yes

4. Is the manuscript presented in an intelligible fashion and written in standard English?

Reviewer #1: Yes

5. Review Comments to the Author

Reviewer #1: Many thanks for the opportunity to review this manuscript. While I think it is an interesting manuscript, I would like to make the following major and/or minor comments:

1. I think the conclusion is we do not know if PR is effective or not, and that we do not know if this pertains to the original studies or to the systematic reviews. Thus, the description of the manuscript should be "toned down".

2. Having created a sensitivity analysis based not least on the most prevalent types of PR would have been helpful.

3. Please specify in every case who has been the reviewers that conducted the literature search, by providing their capital letters.

6. PLOS authors have the option to publish the peer review history of their article (what does this mean?). If published, this will include your full peer review and any attached files.

Reviewer #1: No

---

## [Author Response · Author response to Decision Letter 0]

4 Nov 2023

Dear Felix Bongomin and Reviewers,

Thank you so much for your letter and the reviewer's comments on our manuscript entitled "The role of pulmonary rehabilitation in idiopathic pulmonary fibrosis: an overview of systematic reviews" (ID: PONE-D-23-30793). These comments are very valuable and helpful to revise and improve the academic rigor of our article, as well as the pivotal significance in guiding our other studies. We have carefully studied the comments and made conscientious corrections. Revised portions are marked in red in the manuscript. The main corrections are in the manuscript and the responses to the reviewer's comments are as follows (the replies are highlighted in red).

Responses to the reviewer's comments: 

Reviewer #1:

1. I think the conclusion is we do not know if PR is effective or not, and that we do not know if this pertains to the original studies or to the systematic reviews. Thus, the description of the manuscript should be "toned down". 

Response: We have thought very carefully about the reviewer's question, and we all agree that it is a highly valuable and helpful comment. Based on the studies we included, it might be more reasonable to draw a prudential conclusion that would contribute to the rigor and scholarship of our article. Therefore, we have revised the conclusion according to the reviewer's comments. 

Revised portion: 

Abstract: Conclusion: PR may appear to be an effective and safe treatment for patients with IPF, but the results of this overview should be interpreted dialectically and with caution. Further high-quality, rigorous studies are urgently needed to draw definitive conclusions and provide scientific evidence. (The revision is on page 3, lines 50-54 of the manuscript.)

Conclusion: What is the precise role of PR in the treatment of IPF? Based on the trials we included, definitive conclusions to this question are difficult to draw. It is unclear whether this is related to the quality of the original studies and the variable quality of the evidence on the results (ranging from very low to moderate). Therefore, considering the limitations of the overview, further scientific research is warranted to investigate the efficacy of PR in IPF, for providing a more robust, scientifically rigorous, and accurate foundation for clinical decision-making. (The revision is on pages 20-21, lines 438-445.)

2. Having created a sensitivity analysis based not least on the most prevalent types of PR would have been helpful.

Response: Many thanks to the reviewer for the invaluable and professional advice, this comment is significantly valuable in verifying the robustness of the study findings. We have made our best efforts to correlate prevalent types of PR. Unfortunately, this seems to be challenging given the study type of our article.

The main focus of our study was to assess the risk of bias and evidence quality of the original studies we included using assessment tools to obtain clinical evidence of interest. Presenting a sensitivity analysis based on PR proved difficult as we could not analyze the obtained data quantitatively. And, to our knowledge, there do not appear to be any published overview of systematic reviews-type articles that have used this type of analysis.

We would like to thank the reviewer once again for this comment. This suggestion may have been due to our unclear description of the data synthesis in the article, so we have revised the manuscript (Pages 6-7, lines 145-150, and Page 13, lines 276-277). We have described the data synthesis in the method portion more clearly in the article hoping that this will improve the methodological adequacy and rigor of the article, and we also hope that we can avoid the possibility of misinterpretation by the readers due to the unclear description of the data synthesis methodology.

3. Please specify in every case who has been the reviewers that conducted the literature search, by providing their capital letters.

Response: We apologize for neglecting to detail the researchers who participated in the literature search, and the correction is below.

Revised portion: 

Two researchers (SY Song and WR Liu) independently conducted the literature search (The revision is on page 6, line 134 of the manuscript).

Other changes:

1. Page 1, line 3 and lines 16-17, We have removed the current address of one of the authors.

2. Page 21, lines 457-464, We changed this section to make it more concise and clear, the details of the revision are in the manuscript. 

3. We reformatted the references according to the journal's requirements but did not add or remove any references. 

We did our best to improve the manuscript and ensure that our manuscript met the style requirements of PLOS ONE, so we made some changes to the manuscript. These changes do not affect the content of the paper or the framework. And here we listed the changes and marked them in red in the revised paper.

We appreciate the diligent work of the editor and reviewers. Thank you for your commitment to improving the quality of our academic writing. We hope that all these changes fulfill the requirements to make the manuscript acceptable for publication.

Looking forward to hearing from you soon.

Sincerely,

Jiansheng Li

---

## [Editor Report · Decision Letter 1]

9 Nov 2023

PONE-D-23-30793R1The role of pulmonary rehabilitation in idiopathic pulmonary fibrosis: an overview of systematic reviewsPLOS ONE

Dear Dr. Li,

Thank you for submitting your manuscript to PLOS ONE. After careful consideration, we feel that it has merit but does not fully meet PLOS ONE’s publication criteria as it currently stands. Therefore, we invite you to submit a revised version of the manuscript that addresses the points raised during the review process.

We look forward to receiving your revised manuscript.

Kind regards,

Felix Bongomin, MB ChB, MSc, MMed, FECMM

Academic Editor

PLOS ONE

Journal Requirements:

**Additional Editor Comments:**

Dear authors,

Thank you for revising this work.

However, the results section of the abstract doesn't summarise the goal of these umbrella review.

Are you looking at the quality of the systematic reviews or you want to answer the question on "The role of pulmonary rehabilitation in idiopathic pulmonary fibrosis"?

Kindly extensively revise the abstract.

---

## [Author Response · Author response to Decision Letter 1]

18 Nov 2023

Dear Felix Bongomin,

Thank you so much for your letter and the comments on our manuscript entitled "The role of pulmonary rehabilitation in idiopathic pulmonary fibrosis: an overview of systematic reviews" (ID: PONE-D-23-30793R1). Thank you for your dedication and valuable suggestions on our revised manuscript. We all agree that these comments are essential to strengthen the rigor and readability of our article. Therefore, we have extensively revised the abstract of our article based on your suggestions. Revised sections are highlighted in red in the manuscript. The main corrections in the manuscript are as follows.

Responses to Editor Comments: 

1.Additional Editor Comments: The results section of the abstract doesn't summarise the goal of these umbrella review. ……Kindly extensively revise the abstract.

Response: We have thought deeply about your comments, and we all agree that it is a highly valuable and helpful comment. We apologize for the lack of clarity in the abstract, which may prevent the editor and readers from quickly understanding the main objective of our article through the abstract. Thus, our modifications of the abstract are the following.

Revised portion: 

(1) Background: The role of pulmonary rehabilitation (PR) in idiopathic pulmonary fibrosis (IPF) has been studied in several systematic reviews (SRs), but no definitive conclusions have been drawn due to the wide variation in the quality and outcomes of the studies. And there are no studies to assess the quality of relevant published SRs. This overview aims to determine the effectiveness of PR in patients with IPF and to summarize and critically evaluate the risk of bias, methodological, and evidence quality of SRs on this related topic. (The revision is on page 2, lines 24-30 of the manuscript.)

(2) Results: Seven SRs from 2018-2023 (including 1836 participants) on PR for the treatment of IPF were selected, all of which included patients with a definitive diagnosis of IPF. After strict evaluation by the ROBIS tool and AMSTAR-2 tool, 42.86% of the SRs had a high risk of bias and 85.71% of the SRs had critically low methodological quality in this overview. PR might be effective for patients with IPF on exercise capacity, quality of life, and pulmonary function-related outcomes, but we did not find high quality evidence to confirm the effectiveness. (The revision is on page 2, lines 38-45 of the manuscript.)

(3) We reformatted the references according to the journal's requirements but did not add or remove any references. 

We appreciate your efforts again and hope that the revisions we made will be in line with PLOS ONE’s publication criteria.

Looking forward to hearing from you soon.

Sincerely,

Jiansheng Li

---

## [Editor Report · Decision Letter 2]

21 Nov 2023

The role of pulmonary rehabilitation in idiopathic pulmonary fibrosis: an overview of systematic reviews

PONE-D-23-30793R2

Dear Dr. Li,

We’re pleased to inform you that your manuscript has been judged scientifically suitable for publication and will be formally accepted for publication once it meets all outstanding technical requirements.

Kind regards,

Felix Bongomin, MB ChB, MSc, MMed, FECMM

Academic Editor

PLOS ONE
---

## [Editor Report · Acceptance letter]

13 Dec 2023

PONE-D-23-30793R2 

PLOS ONE

Dear Dr. Li, 

I'm pleased to inform you that your manuscript has been deemed suitable for publication in PLOS ONE. Congratulations! Your manuscript is now being handed over to our production team.

Kind regards, 

on behalf of

Dr. Felix Bongomin 

Academic Editor

PLOS ONE